



# CO₂ Surface Variability, from the Stratosphere or Not?

Michael J. Prather

Earth System Science Department, University of California Irvine, Irvine CA 92617 USA

*Correspondence to*: Michael J. Prather ([mprather@uci.edu](mailto:mprather@uci.edu))

**Abstract.** Fluctuations in atmospheric $CO_2$ can be measured with great precision and are used to identify
human-driven sources as well as natural cycles of ocean and land carbon.  One source of variability is the
stratosphere, where the influx of aged $CO_2$-depleted air can produce fluctuations at the surface.  This
process has been speculated a potential source of interannual variability (IAV) in $CO_2$ that might obscure
the quantification of other sources of IAV.  Given the recent success in demonstrating that the
stratospheric influx of $N_2O$- and chlorofluorocarbon-depleted air is a dominant source of their surface
IAV in the southern hemisphere, we here apply the same model and measurement analysis to $CO_2$.  Using
chemistry-transport modeling or scaling of the observed $N_2O$ variability, we find that the stratosphere-
driven surface variability in $CO_2$ is at most 10% of the observed IAV and is not an important source.  The
southern hemisphere stations with multi-decadal $CO_2$ records can provide clues to sources through the
phase shifts of the IAV relative to the northern hemisphere.


## 1 Introduction

The surface abundance of $CO_2$, a.k.a. the Keeling Curve (Figure 1a), is used as the prime example of the
human-driven increases in greenhouse gases.  It also used to demonstrate control of $CO_2$ by the land
biosphere and the oceans through its annual cycles and interannual variations (Le Quéré et al., 2016;
2018).  The inverse modeling of surface sources based on these $CO_2$ observations is used to infer regional
sources of fossil fuel emissions as well as year-to-year changes in primary productivity of the biosphere
or oceanic degassing (e.g., Gurney et al., 2002; Baker et al., 2006; Engelen et al., 2006; Nassar et al.,
2011; Peylin et al., 2013; Frankenberg et al., 2016; Pandey et al., 2016; Nakazawa, 2020).  There is
concern that atmospheric variations in $CO_2$, and hence the net sources derived from them, may be affected
by interannual variations (IAV) in tropospheric mixing or stratosphere–troposphere exchange (STE)
(Gaubert et al., 2019), but there are no definitive studies.  For example, Nazakawa's (2020) review of
greenhouse gas studies mentions the stratosphere only in connection with $CH_4$ and $N_2O$, not with $CO_2$.

The possibility of a true STE-driven IAV $CO_2$ signal, raised by Gaubert et al. (2019), has not been
seriously investigated.  For the most part, when studies investigate the stratospheric influence on $CO_2$
source inversions, they are not concerned about STE fluxes directly but other factors that degrade the
results:  e.g., gradients across the tropopause, the effective tropospheric air mass diluting surface
emissions, or the inclusion of $CO_2$-depleted stratospheric air in column $CO_2$ calculations (Nassar et al.,
2011; Deng et al., 2015; Frankenberg et al., 2016; Pandey et al., 2016).  For example, Le Quéré et al.
(2018) are concerned how emissions will mix throughout the troposphere and the stratosphere, but not
how stratospheric air will come back down to the surface.  Only studies of the $CO_2$ triple-oxygen isotope
signature ($\Delta^{17}O$) are concerned with accurate STE fluxes, recognizing its importance in the seasonal
isotopic signals (Liang et al., 2017; Koren et al., 2019; Laskar et al., 2019).



Recent work has shown that the stratospheric quasi-biennial oscillation (QBO) modulates the STE and drives much of the IAV observed in surface $N_2O$ through the stratospheric influx of $N_2O$-depleted air (Ruiz et al., 2021; Ruiz and Prather, 2021). It also drives some surface variability in $CFCl_3$ (Ray et al., 2020). Here, we use the $N_2O$ studies of Ruiz et al. (2021) with parallel model simulations of $CO_2$ to place
constraints on the $CO_2$ IAV caused by atmospheric circulation, finding that this effect is a clear but minor perturbation in driving the observed IAV.

## 2 Methods and Analysis

We investigate the $CO_2$ IAV through the historical period 1990-2017 with UC Irvine chemistry-transport model (CTM) simulations and also with surface $N_2O$ observations, which are known to be influenced by STE fluxes.

To study the circulation-driven IAV of $CO_2$, including STE, we focus on the southern hemisphere (SH)
because small fluctuations in the large biosphere-driven seasonality in the northern hemisphere (NH) (Figure 1ab) will obscure any stratosphere-driven IAV. The UCI CTM uses ECMWF Integrated Forecast Fields at 1.1° horizontal resolution and has proven quite successful in simulating the historical IAV of surface $N_2O$, ozone columns, and the Antarctic ozone hole (Ruiz et al., 2021; Ruiz and Prather, 2021; Tang et al., 2021). For $CO_2$, we develop two model scenarios to highlight the impacts of atmospheric
transport. First, we define an emissions-driven, eCO2 scenario, in which the total atmosphere increases at a constant rate of 2 ppm (parts per million, dry-air mole fraction) $y^{-1}$, forced by uniform surface emission over all land and ocean areas from 20° N – 60° N. This eCO2 scenario is a simple experiment to test how atmospheric circulation driving a NH-SH gradient might affect the seasonal and interannual variability of SH surface $CO_2$; it is obviously not realistic, lacking the large NH seasonality. A second, stratospheric-
driven, sCO2 scenario, is forced with a net stratospheric flux of $CO_2$-depleted air being transported into the troposphere and down to the surface. This STE flux is calculated as the equivalent of the aging of stratospheric $CO_2$ relative to the troposphere (2 ppm $y^{-1}$), yielding an apparent negative $CO_2$ flux of 0.8 PgC $y^{-1}$. This forcing flux is placed in the uppermost model layer (~80 km altitude) and transported to the surface. In both of these cases, $CO_2$ is changing linearly with a known trend, and we subtract that trend to
get the modeled anomalies. A third, independent method uses the observed SH surface $N_2O$ signal, driven by stratospheric photochemical loss of 13 TgN (as $N_2O$) $y^{-1}$, as a measure of STE influence. In this case we scale the results to $CO_2$ using the ratio of the STE fluxes, i.e., 0.15 ppm $CO_2$ per ppb $N_2O$.

The monthly $CO_2$ surface observations are gathered from NOAA ESRL (Dlugokencky et al., 2021a). We
use 5 sites: BRW = Barrow AK, 71° N, 156° W; MLO = Mauna Loa HI, 20° N, 156° W; SMO = Tutuila, Am. Samoa, 14° S, 171° W; CGO = Cape Grim, Tasmania, Australia, 41° S, 144° E; and SPO = South Pole, 90° S. Monthly average in situ observations are used with gaps filled by flask data at the same site. CGO is flask only. We have a continuous monthly record from 1985 through 2020 (Figure 1a). We convert these to a stationary series of residuals by fitting polynomials: using 2nd, 3rd, 4th, and 5th order
polynomials produced are almost identical results for any site (not shown), and we averaged the residuals from the 3rd and 4th order fits. Results for the average of SPO+CGO are shown in Figure 1c as the red line, which shows a clear annual cycle plus equally large variability on decadal scales. The annual cycle (min to max) in $CO_2$ and its rate of change is a critical metric used to evaluate the carbon cycle in Earth system models (Graven et al., 2013; Zhao and Zeng, 2014; Wenzel et al., 2016). Here, we calculate the
cycle simply by averaging each calendar month of the year as shown in Figure 1b. The annual amplitudes (min to max) are 16.4, 6.4, 0.92, 1.02, and 1.14 ppm for BRW, MLO, SMO, CGO, and SPO, respectively.



These are consistent with those previous studies, although SH cycles remain understudied and not well evaluated. In some months SMO at 14°S can be north of the South Pacific Convergence Zone and thus influenced by NH air, explaining its non-sinusoidal cycle when compared with SPO and CGO. Also

shown is the annual cycle for the modeled eCO2 scenario (~0.18 ppm, dotted lines for SMO and SPO), while that for the sCO2 scenario is even smaller (~0.06 ppm) and not shown. It is interesting that the SH annual cycles in eCO2 are similar in shape to those observed, even catching the double peak at SMO, but the magnitude is much smaller. There is no evidence in our direct modeling or analysis that stratosphere-troposphere exchange, which does have an annual cycle in $N_2O$, can produce a detectable annual cycle in

$CO_2$ above the large observed cycle.

Because of the clearly observed QBO signal in SH surface $N_2O$ (Ruiz et al., 2021), we look for a $CO_2$ IAV signal in the 2–5 year range, removing the large 6–10 year IAV with a high-pass filter after removing the annual cycle with a low-pass filter (see Figure 1c for details). The resulting SPO+CGO

IAV signal is shown as a thick black line in Figures 1c & d. In the SH where the annual cycle is small, the low-pass filter clearly removes any vestige of an annual cycle; but in the NH, especially BRW, the annual cycle is large and slightly irregular in amplitude and thus the low-pass filter is unable to remove all of it.

For monthly $N_2O$ surface observations, also from NOAA ESRL (Dlugokencky et al., 2021b), we focus only on SH mid- and polar latitudes, using SPO, CGO, plus 3 other sites: S30 = Western Pacific Cruise, 30° S, 168° E; USH = Tierra del Fuego, Ushuaia, Argentina, 55° S, 68° W; and PSA = Palmer Station, Antarctica, 65° S, 64 °W. All five sites have nearly identical $N_2O$ records, and we average them to get our SH IAV signal with the same processing as for $CO_2$. The QBO circulation is known to reach

throughout the stratosphere and into the troposphere (Tung and Yang, 1994; Hamilton and Fan, 2000), and multi-model studies have attributed the $N_2O$ IAV to the modeled STE flux (Ruiz et al., 2021). We can thus scale the surface $N_2O$ IAV with the ratio of STE fluxes ($CO_2$:$N_2O$) to give an observational estimate of the STE-driven IAV in $CO_2$ (dashed blue line in Figure 1d). The IAV in SH (40° S – 90° S) surface $CO_2$ calculated from the sCO2 model scenario is also shown (dashed red line in Figure 1d). The

modeled sCO2 and observed $N_2O$-scaled IAVs are not always phase, but they are in strong agreement in terms of amplitude: the STE IAV in $CO_2$ is small, at most 10% of the observed IAV in the 2-5 year range. Similarly, the modeled eCO2 IAV shows that tropospheric circulation changes do generate IAV, but these are at most 12% of that observed.

We compare $CO_2$ with other interannual cycles in the Earth system by plotting in Figure 1d (i) the QBO phase change (from easterly to westerly zonal equatorial wind at 40 hPa, as thick gray vertical bars, see Newman, 2021) and (ii) the times of moderate to extreme El Niños (red stars) and La Niñas (blue stars) (Trenberth, 2021). From this analysis, we expect minimal contribution of the QBO–driven circulation to the $CO_2$ IAV, and find no obvious additional correlation between the two. For the El Niño–Southern

Oscillation (ENSO), this simple comparison is inadequate. At best it shows that the larger positive SH IAV follows the El Niños; whereas we know that ENSO affects ocean upwelling and continental rainfall and the $CO_2$ anomalies correlate very well with tropical ocean temperatures (Wang et al., 2021; Keeling and Graven, 2021).

**3. Conclusions, Speculations, and Digressions**



We have shown that the STE fluxes of old stratospheric air with "depleted" $CO_2$ have little influence on the IAV or annual cycle. Similarly, the atmospheric transport of NH fossil fuel emissions into the SH produces signals that are 10–15% of what is observed. Thus the speculations of Gaubert et al. (2016) regarding atmospheric transport can be dismissed.

The latitudinal pattern of $N_2O$ IAV provides evidence for causes: e.g., the STE-driven signal weakens in the tropics and changes phase in the NH; QBO composites show a clear separation of hemispheric sources (Ruiz et al., 2021). For $CO_2$ the 2–5 year IAV has similar amplitude but some phase shifts within the SH (SPO+CGO vs. SMO, solid and dotted black lines in Figure 1d). To expand on this, we reprocessed the monthly $CO_2$ data with a simpler algorithm to avoid possibly spurious signals from the band-pass filtering. We recognize that many options are available (see review in Piao et al., 2020). First, a polynomial fit to each series as defined above is removed. With 432 equally-weighted monthly values, the polynomial fit is extremely well determined and gives residuals and a stationary time series like the red curve in Figure 1c. Second, a 12-month running mean removes most of the annual cycle and gives 421 monthly values from 1 July 1985 to 1 July 2020. This processing leaves the 2–10 year cycles, and the resulting IAVs for MLO, SMO, CGO, and SPO are shown in Figure 1e. The BRW signal is very noisy and not shown here. At first glance, the sites are similar. This agreement is expected because most of this signal occurs over time scales >2 years and the interhemispheric mixing time scale is ~1 year. On time scales <2 years we find many instances where the amplitude is distinct by hemisphere or the phase is shifted by 6–12 months (the interhemispheric mixing time). When the four sites are in synch, one can only presume that the $CO_2$ perturbation is tied to changes in the growth/decay of tropical biomass transported equally to both hemispheres (Keeling and Graven, 2021). When the IAV change is opposite (e.g., years 2008, 2010, 2018) the $CO_2$ perturbation likely occurred distinctly in one hemisphere, but it does not identify which. In some periods the NH obviously leads (e.g., 1993, 1998, 2003) or lags (e.g., 1991, 2002, 2015) the SH by about a year, again implying a single hemispheric origin, but now identifying the hemisphere because the global multi-year trend is clear.

The Samoan site SMO provides a valuable but very challenging record of $CO_2$ and other trace gases with dominant NH emissions (e.g., $N_2O$, $SF_6$, chlorofluorocarbons). As seen in Figures 1de, sometimes SMO is synchronous with the SH mid- and polar-latitude sites (CGO and SPO, which are almost always synchronous with each other) and at other times it links with MLO and the NH. Thus, to use SMO $CO_2$ as a metric for carbon cycle models, one must recognize that SMO is not representative of the SH tropics. When evaluating carbon cycle models, one must be sure that the tracer transport in these models has the correct IAV for SMO. On interannual time scales there are clear times when SMO aligns as SH and is distinct from MLO (e.g., 1986, 2002, 2010, 2018) and times when it aligns with MLO and is distinct from CGO+SPO (e.g., 1999, 2012, 2015, 2017). These interannual shifts provide an excellent test for $CO_2$ historical simulations using weather forecasting systems (e.g., McNorton et al., 2020) and realistic sources and sinks (e.g., Piao et al., 2018; Wang et al., 2020).

A by-product of this alternate method is a cleaner, statistical measure of the annual cycle. Each of the 421 running-mean quantities in Figure 1e consists of 12 months, and each contains an annual cycle. If the cycle is sinusoidal, then the min-to-max amplitude is equal to twice the square root of 2 times the standard deviation. With this approximation, we calculate a mean amplitude of 17.0, 6.4, 1.4, 1.2, and 1.2 ppm for BRW, MLO, SMO, CGO, and SPO, respectively. These results are almost identical to those from the composited annual cycles plotted above, but disagree at SMO as might be expected because of its double-peaked annual cycle. A linear fit to the standard deviations gives trends for the period 1985-



2020 of 0.11, 0.014, 0.008, 0.008, and 0.003 ppm y$^{-1}$ for BRW, MLO, SMO, CGO, and SPO, respectively. In terms of % y$^{-1}$ the increases in annual cycle amplitude are 0.62, 0.22, 0.62, 0.67, and
0.25. Are these trends significant? Without trying to assess the number of degrees of freedom in the time series, we simply compare the 36-year change in annual cycle amplitude (signal) with the standard deviation of the residuals after the fit (noise). This ratio is about 4:1 at BRW, making the change in annual cycle obvious to the eye; the ratio is 1:1 for MLO, SMO and CGO, making for a solid detection; but the ratio at SPO is much less because the trend in annual amplitude is much smaller. Our results for
BRW and MLO agree well with other more extensive data analyses (Graven et al., 2013; Zhao and Zeng, 2014; Wenzel et al., 2016; Piao et al., 2018; Wang et al., 2020), but provide subtler tests of the seasonality and trends in the SH, which is not often used for model evaluation. A more serious uncertainty analysis focusing on the SH sources and sinks, the annual and IAV cycles, and their trends would help solidify our knowledge of the carbon cycle.

**Code availability**. The full chemistry-transport model (CTM) code used for the tracer simulations is included in the archived data.

**Data availability**. All data and code used in this analysis are placed in the archive at datadryad.org. The CTM code is in FORTRAN, and the post analysis code is in Matlab.

**Competing interests**: The author declares that they have no conflict of interest.

**Acknowledgements**. The author acknowledges the support and assistance of his research group at UC Irvine, particularly Xin Zhu for the CTM simulations and Daniel Ruiz for the N$_2$O simulations.

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

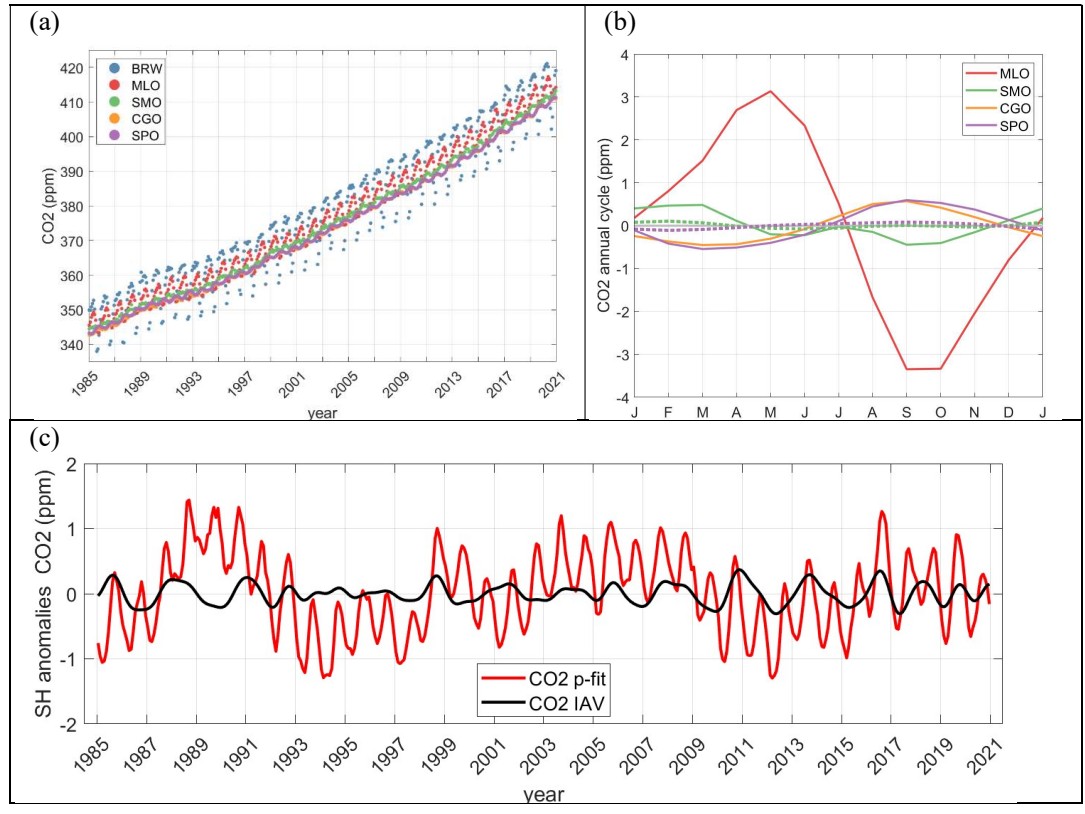


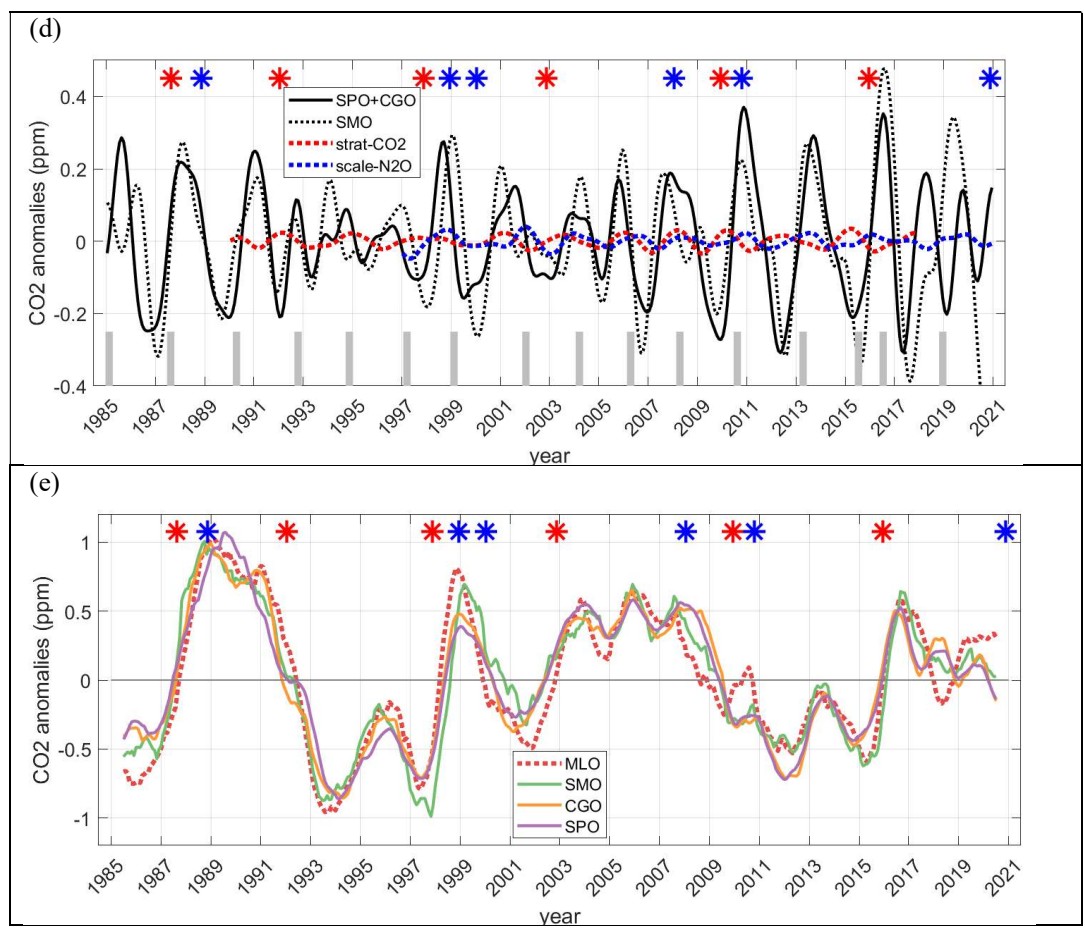

**Figure 1 (a):** NOAA surface $CO_2$ monthly data (ppm, mole fraction) from Dlugokencky et al. (2021a). The 5 sites are: BRW = Barrow AK, 71°N, 156°W; MLO = Mauna Loa HI, 20°N, 156°W; SMO = Tutuila, Am. Samoa, 14°S, 171°W; CGO = Cape Grim, Tasmania, Australia, 41°S, 144°E; SPO = South Pole, 90°S. Monthly average in situ observations are used with gaps filled by flask data at the same site. CGO is flask only.

**(b):** Mean annual cycle in surface $CO_2$ (ppm) at 4 sites (BRW not shown) using 1985 through 2021. Amplitudes (max minus min) are 16.4 ppm (BRW), 6.4 (MLO), 0.92 (SMO), 1.02 (CGO), 1.14 (SPO). Zonal-averaged results from the UCI CTM using the emissions-driven scenario (eCO2) are shown as dotted lines for SMO and SPO (same color coding for each site) The annual cycle in eCO2 has similar phasing at SPO and is double-peaked at SMO, but the amplitudes (~0.18 ppm) are much smaller and barely detectable on this scale. The stratosphere driven scenario (sCO2) has an even smaller amplitude (~0.06 ppm). The atmosphere-only transport model uses historical meteorology from the ECMWF IFS system. The eCO2 scenario is driven with a uniform surface flux from 20°N to 60°N equivalent to 2 ppm annual increase in $CO_2$ (~4.2 PgC/y). The sCO2 scenario is driven by a negative $CO_2$ flux from the mesosphere that is equivalent to 2 ppm/y loss over the stratosphere only (~0.8 PgC/y).

**(c):** Residual surface $CO_2$ variability (ppm) averaged over two primary southern extra-tropical stations (SPO+CGO, red line) after removal of a polynomial fit (average of 3rd and 4th order fits). Polynomial fits of 3rd, 4th, and 5th-order produce nearly identical $CO_2$ residuals. SMO, a tropical station, is not used because it has very different annual cycles, see (b) above. This residual curve shows evidence of annual to decadal variability. The interannual variability (IAV, black line) is calculated by a sequence of high-pass (Matlab: *highpass(series, 0.32, 12)*) followed by low-pass (Matlab: *lowpass(series, 0.30, 12)* filters leaving variability with periods of 2-5 years to focus on the stratospheric quasi-biennial oscillation (QBO). The low-pass filter results for these SH sites are almost identical to a 12-month running mean series.



345
**(d):** The IAV of surface $CO_2$ for SH mid-latitudes (black solid line from Fig, 1c) is replotted on an expanded scale and is compared with that modeled by the UCI CTM using the stratospheric driven $CO_2$ scenario ('strat-CO2', dashed red line) and with the corresponding SH IAV observed in surface $N_2O$ that has been scaled to match the equivalent $CO_2$ fluxes ('scale-N2O', dashed blue line) at 0.15 ppm $CO_2$ per ppb $N_2O$. The SH extra-tropical IAV is the average of SPO+CGO. SMO (shown as dashed black

350 line) has a very similar IAV to that of SPO+CGO in spite of the SMO having a very different annual cycle. The timing of the QBO phase change in equatorial zonal wind at 40 hPa from negative (easterlies) to positive (westerlies) is denoted with thick vertical gray bars. The timing of moderate to extreme El Ninos (red stars) and La Ninas (blue stars) are also shown.

**(e):** The IAV of surface $CO_2$ for 4 sites (SPO, CGO, SMO, MLO) emphasizing the 2-10 yr IAV. The residuals are calculated
355 separately for each site by removing the average of the $3^{rd}$ and $4^{th}$-order polynomial fits and then performing a 12-month running mean. Every month is weighted equally. The timing of moderate to extreme El Ninos (red stars) and La Ninas (blue stars) are also shown.