# Peer review of "CO2 Surface Variability, from the Stratosphere or Not?"

_Earth System Dynamics, 2021_

## Referee Comment (RC2)

This is concise analysis of the potential impact of stratospheric variability on $CO_2$ mixing ratios at the surface, using similar methods to those used in Ruiz et al. (2021) to analyze $N_2O$. The expected variability for $CO_2$ at the surface from the stratosphere is found here to be small compared to the actual observed interannual variability (IAV), suggesting that this effect is not as important as in the case of those other gases. Two methods to quantify the effect were used: modeling the stratospheric effect using a full 3D transport model, and the stratospheric effect as inferred from actual $N_2O$ variability with a scaling factor used to convert to $CO_2$. These methods give a peak-to-peak amplitude of about 0.05 ppm in the Southern Hemisphere, which is small compared to a value of about 0.5 ppm, ten times that, that they compute from actual monthly time series at South Pole (SPO) and Cape Grim (CGO), as well as Samoa (SMO).

This is a nice paper and would be good to publish, to put to rest speculation that this effect may be large enough to matter much, compared to the other drivers of $CO_2$ variability. However, I have a concern that the analysis of the $CO_2$ data at CGO and SPO (and SMO, too, I suppose, though I did not check it) has not been done correctly: in particular, the calculation of the black 'IAV' curve in Figures 1c and 1d. When I attempted to do the same calculation in MATLAB, I was able to replicate the red curve in Fig 1c, but I get a very different curve than the black one when I filter out Fourier components with periods longer than 5 years and shorter than two years. Please see my figure below that shows what I get for that black curve (I used 5.5 and 1.5 years as the cutoff). The amplitude of the variability is larger, on the order of 0.8 ppm, and the location of the peaks is completely different that what Prather obtains. I found the MATLAB *lowpass* function difficult to work with (in that the cutoff frequency did not seem to correspond to the results obtained), but the *highpass* function seems to work. I used it to quantify the portion of the signal corresponding to *highpass(signal,0.182,12)* and *highpass(signal,0.67,12)*, or those frequencies higher than cutoffs corresponding to periods of 5.5 and 1.5 years, respectively, and subtracted the second from the first. Prather says in the caption to Figure 1c that he ran the detrended timeseries through a *highpass(series,0.32,12)* filter followed by a *lowpass(series,0.3,12)* filter. According to my understanding of these functions, that corresponds to keeping all periods shorter than 1/0.32 = 3.125 years, then those longer than 1/0.3 = 3.333 years, in which case no frequencies should be left. While I was not able to understand how the given frequency cutoff functioned in the case of *lowpass*, I did figure it out in the case of *highpass*, and the description in the caption does not seen to agree with inclusion of periods from 5 to 2 years. The black curve given in Figs 1cd appears of show beat phenomena (with an expanding and contracting envelope inside of which the variability occurs) that I don't see in my results. I would ask Dr. Prather to check what he has done here and get back to me before proceeding towards publication (i.e. I'd like another look at that point). Perhaps the curve I get might correspond more closely with the timing of the QBO phases given at the bottom of Figure 1d -- I thought I might have detected some coincidence for some of the larger fluctuations. If the black curve in Figs 1cd changes, this could impact the discussion in the text.

[Figure]

Figure: Filtered CO$_2$ variability for periods between 1.5 and 5.5 years, based on gap-filled monthly NOAA data at CGO and SPO (averaged).

The paper is well-written; I only had a few editorial comments, below:

61: add "there" at end of sentence, for clarity.

335: add "a" before "2 ppm"

Fig 1: Does the NOAA analysis remove any frequencies in the gap-filling calculation that they do? Do they attempt to make the 'months' equally-spaced? (i.e. to remove the impact of the differing month lenghts, February in particular -- I believe they do do this for some of their products)

66: if the emissions are spatially uniform, how can you generate a N/S gradient? Are you referring to the creation of a N/S gradient from the flat emissions field due to transport effects?

85: remove "are"

99: instead of "does have", say "does drive" or "does cause"?

138: add "of CO$_2$ at the surface" after "annual cycle".

139: what sort of signals -- IAV? If so, please say that... for clarity.

---

## Author Response (AR1)

I thank the reviewers for their recognition that a null result can be worth publishing. Further, I am indebted to their efforts in digging into this manuscript, asking tough questions, and even trying to reproduce the results. I spent the last few weeks learning the vagaries of frequency filtering to derive the interannual variability (IAV) with periods between 1 and 5 years (see response to RC2). I am now more confident that the new IAV with band-pass filtering is solid, at least for the middle of the observing period. The comparison in revised Fig 1c between the band-pass and the independent method using 12-month running means is close enough. Fortunately, the change in frequency filtering, brought about by Reviewer 2, did not change any of the major conclusions because the time series being compared were all processed with the same filtering.

Michael Prather

This revised manuscript, I believe, addresses all of the issues raised by both reviewers. The new test is shown in **red**. Major changes are: (i) the rewrite of the methods section to describe more carefully the new, better documented band-pass filtering; (ii) the discussion of old Fig 1e now dropped with much of the paragraph in Lines 141-162: (iii) new Fig 1e showing trends with standard error in annual amplitude; and (iv) shortened figure (not highlighted for changes).

*RC1*

> *This is an interesting paper addressing an oft-overlooked influence on trace gas abundances at Earth's surface. The results, while indicating no significant influence on interannual variations in atmospheric CO2 mole fractions, is important for ensuring that interpretations of the CO2 atmospheric record are free from significant bias. I only have a few suggestions for the author to consider that I believe, if addressed, would improve the robustness of the conclusions and the appropriateness for this work to be published in this Journal.*
>
> *Isn't it true that the uniform emission scenario simulation should also result in the development of persistent trop-strat gradients in CO2 mole fraction? As a result, the calculated surface mole fractions should include some effect of STE variability, not just tropospheric transport and mixing. So I'm surprised that the results described as investigating tropospheric transport IAV only (**lines 65-70, lines 122-123**), and that the results are recast as indicating only the influence of atmospheric transport of NH fossil fuel emissions into the SH (**line 138-139**). Consider a more accurate description of this scenario.*

Yes, this is correct; sorry for the confusion. With NH emissions at a rate of 2 ppm/year, a strat-trop gradient will reach steady state and the SH IAV will reflect variability in STE as well as interhemispheric transport. Those sections are revised with suitable caveats. The eCO2 simulation still emphasizes the interhemispheric IAV because the SH STE flux is ~0.4 PgC/yr (sCO2) while the interhemispheric mean flux is ~2 PgC/yr (eCO2) (equivalent to increasing SH troposphere by 2 ppm/yr).

Lines 65-70 (and later in the paragraph) have been revised to note this, and to compare the relative fluxes. Lines 122-123 do not then need to be revised with these caveats. The full IAV from fossil fuels would have to be calculated with a more complete C-cycle model (i.e., full fossil fuel emissions of ~4 ppm/yr plus land and ocean sinks in the correct locations and with the correct seasonality. That would be interesting, but not here.

*The sCO2 scenario, designed to isolate the influence of interannual variability in strat-trop exchange, is set up by including a negative forcing flux at 80 km (top of the model), which then gets transported to the surface. This simulation and a rescaling of results for N2O to make it more appropriate for CO2 are used as an indication of the expected influence of STE on surface CO2 mole fractions. While the sCO2 and rescaled N2O flux results have a similarly small amplitude (a main finding of the paper), they are not often in-phase, suggesting issues with the usefulness (accuracy?) of one or the other as tools for investigating the processes in question here.*

Correctly described. The not-so-perfect phasing of the surface signal from the two methods is primarily due to the modeled signal (sCO2) not producing a perfect phasing match with the observed surface N2O, although the amplitude is well matched (see the Ruiz papers: 2021, 2022). With the correction of the band-pass filtering identified by Reviewer #2, the phasing of sCO2 and scaled-N2O (revised Figure 1d) are more similar, but still not a great match.

*To make the conclusion more robust, the author might consider (show) whether or not the results (phasing of variability and amplitude in the sCO2 results at the surface) are influenced by where in the stratosphere the growth rate-related flux (-0.8 pgC/yr) is placed in the model (currently at 80km). Related results for the phasing of CFCs and other gases (STE influence at Earth's surface) has shown that it is influenced by the altitude where the loss flux occurs in the stratosphere. As a result, I think it is important to demonstrate that placing the CO2-related loss flux into a lower model level doesn't affect the conclusions being made here (i.e., the timing of the phasing in the resulting surface mole fraction IAV for CO2, or its amplitude).*

An important result from the Ruiz studies is that the STE fluxes calculated for sCO2, N2O, and even CFC-11 are all in phase, and they scale with the annual mean stratospheric flux driving them. We find, using both the GMI and UCI chemistry-transport models, that the STE flux (seasonal and IAV) scales perfectly with the total net loss of CFC-11 and N2O in the stratosphere, even though CFC-11 has twice the loss and it occurs almost a scale height lower than that of N2O. In other words, the STE fluxes are driven by variability in the lowermost stratosphere (altitudes below ~16 km) and not by the losses in the tropical mid stratosphere. We expanded upon this at the end of the paragraph (Line 77).

*The author also includes some conclusions on the causes of interannual variability in CO2 over 2 to 10 year time spans. But I'm not convinced that the periods identified by the author in which the NH or SH lead are convincing or obvious. Perhaps, given the difficulty in interpreting the SMO results, as discussed, the figure 1e should only include three sites? A shortcoming of using MLO in the NH is that it too can be influenced by unusual transport regimes (like SMO). Are there other NH sites not made overly noisy by the large photosynthetic uptake signal that might also be considered here to add to the robustness of this analysis?*

Looking hard at this Figure 1e, I am less confident of some of my conclusions. One advantage of that figure is that the signal processing is simple and does not induce any of the possible artefacts or edge effects that band pass filtering does (i.e., simple polyfit followed by 12-month running mean). Nevertheless, the figure does not make the case for SMO-CGO separation any better than does Figure 1d. So it is replaced by new figure you suggest below.

*Finally, the analysis of seasonal amplitudes in CO2 annual cycles is interesting, but currently lacks some indication of the goodness of fit of the linear regression. Can a figure showing the amplitudes derived from the 12-month standard deviations be shown over time? Or some appropriate metrics be cited?*

[Figure]

*Details:*

*Line 18-19, last sentence of abstract isn't very clear.*

This sentence was based on discussion of Figure 1e. The figure is now replaced by one showing the annual amplitudes, and the new abstract sentence reflects that figure.

*Line 24-25, verb missing*

Yes, corrected: inserted "is".

*Line 46-48, line 101. inaccurate representation of the Ray et al. paper, which demonstrated the particular effect being discussed here for N2O, CFC-11 and CFC-12, not just CFC-11. And that paper also focused on results from the SH.*

Yes, that section has been revised to reflect Ray and earlier works on $N_2O$ variability from the stratosphere that predates either of these recent papers.

The Ray reference is not included in Line 101, because the Ray paper only reported SH averages and did not define the QBO pattern in terms of latitude-time. The opening sentence there has been revised to clarify this.

*Line 66, emissions are uniform in space \*and constant over time\*?*

Yes.  Fixed.

*Line 85, typo*

Sorry about that lack of proofing.  The clause is revised to:  "The 2nd, 3rd, 4th, and 5th order polynomials produced almost identical results for any site …

*Line 139, 2019 not 2016*

Corrected, thanks.

*Line 148, unusual wording—fit was removed, but what was retained?*

This has been revised and corrected. with the new 'methods' discussion asked for re Figure 1b below.

*Lines 165-170, the unusual nature of SMO has been demonstrated for other long-lived trace gases, some citation to those studies seems warranted here to show that it isn't just an issue for CO2.*

Good point, a sentence has been added noting that SMO's ambiguous NH-SH identity was first clearly seen in the chlorofluorocarbons (Cunnold et al., 1994) and is well studied for N2O (Nevison et al., 2007).

*Methods and results are included in the Figure 1b caption—likely better placed elsewhere. In fact I think most (all?) of that info already is in the main text.*

Agreed, I have checked/revised the main text and eliminated most of the redundant information from the caption.

New references:

*Cunnold, D. M. Fraser, P. J. Weiss, R. F. Prinn, R. G. Simmonds, P. G. Miller, B. R. Alyea, F. N. and Crawford, A. J.:  Global trends and annual releases of CCl₃F and CCl₂F₂ estimated from ALE/GAGE and other measurements from July 1978 to June 1991, J. Geophys. Res.,99(D1), 1107-1126, doi: 10.1029/93JD02715, 1994.*

*Nevison, C. D., Mahowald, N. M. Weiss, R. F., and Prinn, R.G.:  Interannual and seasonal variability in atmospheric N₂O, Global Biogeochem. Cycles, 21, GB3017, doi:10.1029/2006GB002755, 2007.*

*Nevison, C. D., Kinnison, D. E., and Weiss, R. F.: Stratospheric influences on the tropospheric seasonal cycles of nitrous oxide and chlorofluorocarbons, Geophys. Res. Lett., 31, L20103, https://doi.org/10.1029/2004gl020398, 2004.*

*Nevison, C. D., Dlugokencky, E., Dutton, G., Elkins, J. W., Fraser, P., Hall, B., Krummel, P. B., Langenfelds, R. L., O'Doherty, S., Prinn, R. G., Steele, L. P., and Weiss, R. F.: Exploring causes of interannual variability in the seasonal cycles of tropospheric nitrous oxide, Atmos. Chem. Phys., 11, 3713–3730, https://doi.org/10.5194/acp-11-3713-2011, 2011.*

*RC2*

*This is concise analysis of the potential impact of stratospheric variability on CO2 mixing ratios at the surface, using similar methods to those used in Ruiz et al. (2021) to analyze N2O. The expected variability for CO2 at the surface from the stratosphere is found here to be small compared to the actual observed interannual variability (IAV), suggesting that this effect is not as important as in the case of those other gases. Two methods to quantify the effect were used: modeling the stratospheric effect using a full 3D transport model, and the stratospheric effect as inferred from actual N2O variability with a scaling factor used to convert to CO2. These methods give a peak-to-peak amplitude of about 0.05 ppm in the Southern Hemisphere, which is small compared to a value of about 0.5 ppm, ten times that, that they compute from actual monthly time series at South Pole (SPO) and Cape Grim (CGO), as well as Samoa (SMO).*

*This is a nice paper and would be good to publish, to put to rest speculation that this effect may be large enough to matter much, compared to the other drivers of CO2 variability.*

Thank you.

*However, I have a concern that the analysis of the CO2 data at CGO and SPO (and SMO, too, I suppose, though I did not check it) has not been done correctly: in particular, the calculation of the black 'IAV' curve in Figures 1c and 1d. When I attempted to do the same calculation in MATLAB, I was able to replicate the red curve in Fig 1c, but I get a very different curve than the black one when I filter out Fourier components with periods longer than 5 years and shorter than two years. Please see my figure below that shows what I get for that black curve (I used 5.5 and 1.5 years as the cutoff). The amplitude of the variability is larger, on the order of 0.8 ppm, and the location of the peaks is completely different than what Prather obtains. I found the MATLAB lowpass function difficult to work with (in that the cutoff frequency did not seem to correspond to the results obtained), but the highpass function seems to work.*

*I used it to quantify the portion of the signal corresponding to highpass(signal,0.182,12) and highpass(signal,0.67,12), or those frequencies higher than cutoffs corresponding to periods of 5.5 and 1.5 years, respectively, and subtracted the second from the first. Prather says in the caption to Figure 1c that he ran the detrended timeseries through a highpass(series,0.32,12) filter followed by a lowpass(series,0.3,12) filter. According to my understanding of these functions, that corresponds to keeping all periods shorter than 1/0.32 = 3.125 years, then those longer than 1/0.3 = 3.333 years, in which case no frequencies should be left. While I was not able to understand how the given frequency cutoff functioned in the case of lowpass, I did figure it out in the case of highpass, and the description in the caption does not seem to agree with inclusion of periods from 5 to 2 years. The black curve given in Figs 1cd appears of show beat phenomena (with an expanding and contracting envelope inside of which the variability occurs) that I don't see in my results. I would ask Dr. Prather to check what he has done here and get back to me before proceeding towards publication (i.e. I'd like another look at that point). Perhaps the curve I get might correspond more closely with the timing of the QBO phases given at the bottom of Figure 1d -- I thought I might have detected some coincidence for some of the larger fluctuations. If the black curve in Figs 1cd changes, this could impact the discussion in the text.*

***Thank you very much.*** I am truly indebted to you for taking the effort to check on this. I indeed also found the Matlab *lowpass* function to be illogical in terms of the frequencies filtered. I had simply tuned it with a range of cutoff frequency parameters using a synthetic spectrum with frequencies 1/1, 1/2.3, and 1/6 per year. That was a mistake.

You inspired me to redo the entire processing and understand the 'artistry' sometimes involved with frequency filtering of these data. First, I checked and was able to reproduce your figure (top) by subtracting the two highpass filters as you suggest (my figure bottom). The differences in the early years may have to do with the CGO+SPO merge and gap fill in the early years, and that I truncated the observations pre-1985. I think my figure below is close enough. Of course you were correct, the IAV is much larger than in the high/lowpass combination I used in the earlier manuscript.

[Figure]

I then investigated a range of filtering options and settled on the Matlab function '*bandpass*', because it seemed the most straightforward (it passes through information within a bandpass range), and it was the most well documented:

        https://www.mathworks.com/help/signal/ref/bandpass.html

For the monthly times series it is called with the sequence below, including a backwards call, to avoid errors and phase shifts at the edges. (Related to the filtfilt in matlab.)

```
co2bp1 = bandpass(co2resid, [0.2 0.8], 12, 'ImpulseResponse','iir', 'Steepness',0.85);
co2bp2 = flip(bandpass(flip(co2resid), [0.20 0.80], 12, 'ImpulseResponse','iir', 'Steepness',0.85));
co2bp = 0.5*(co2bp1+co2bp2);
```

My figure below shows the two methods: hp-hp is yours, and bp+bpf is now being used. For most of the time series they agree very well, but at the edges, all frequency filtering has problems. Basically I would not trust the edges ±2 years. This is important because it implies that it is almost impossible to assess growth changes or IAVs in the last two years. I also looked at the IAVs for a range of lower and upper band pass limits, and settled on [0.2 0.8] as being the best central values that caused the least spurious fluctuations at the edges. All data analysis in the manuscript was redone with *bandpass*.

[Figure]

*The paper is well-written; I only had a few editorial comments, below:*

*61: add "there" at end of sentence, for clarity.*

Agreed, that is much better.

*335: add "a" before "2 ppm"*

Corrected, although the duplication of the methods, here and in the main text, has some of this caption dropped.

*Fig 1: Does the NOAA analysis remove any frequencies in the gap-filling calculation that they do? Do they attempt to make the 'months' equally-spaced? (i.e. to remove the impact of the differing month lengths, February in particular -- I believe they do do this for some of their products)*

The NOAA data used here are not particularly processed. I have taken the monthly means directly from the measurements as reported. All monthly points are treated as equal in the times series. That might slightly distort the annual cycle, but probably not the IAV. The only gap-filling here, where most data are monthly means from continuous measurements, is the use of "monthly" flask data (whatever the monthly means). (Also, May 2015 SMO had to be interpolated.) In a previous study centered on N2O (Ruiz et al., 2021; 2022), we have used the NOAA-filtered and NOAA-processed observations that removed the annual cycle and trend. But here, we use only the monthly unprocessed values for CO2 and N2O from the web site.

*66: if the emissions are spatially uniform, how can you generate a N/S gradient? Are you referring to the creation of a N/S gradient from the flat emissions field due to transport effects?*

This was confusing because the last clause (i.e., emissions are only from the surface at 20N-60N) does not read well. That clause has been revised, clearly stating the methods.

*85: remove "are"*

Yes. That clause has been further revised for clarity (noted by both reviewers).

*99: instead of "does have", say "does drive" or "does cause"?*

Agreed, have chosen "does drive an annual cycle…"

*138: add "of CO2 at the surface" after "annual cycle".*

Yes, that is better.  Thanks.

*139: what sort of signals -- IAV? If so, please say that... for clarity.*

Agreed.  This opening paragraph has been revised to give specific numbers and comparisons.  It is more direct now.